# Enzymatic one-step ring contraction for quinolone biosynthesis

Shinji Kishimoto[1], Kodai Hara[1], Hiroshi Hashimoto[1], Yuichiro Hirayama[1], Pier Alexandre Champagne [2], Kendall N. Houk[2,3], Yi Tang [2,3] & Kenji Watanabe[1]

The 6,6-quinolone scaffolds on which viridicatin-type fungal alkaloids are built are frequently found in metabolites that display useful biological activities. Here we report in vitro and computational analyses leading to the discovery of a hemocyanin-like protein Asql from the *Aspergillus nidulans* aspoquinolone biosynthetic pathway that forms viridicatins via a conversion of the cyclopenin-type 6,7-bicyclic system into the viridicatin-type 6,6-bicyclic core through elimination of carbon dioxide and methylamine through methyl isocyanate.

[1] Department of Pharmaceutical Sciences, University of Shizuoka, Shizuoka 422-8526, Japan. [2] Department of Chemistry and Biochemistry, University of California, Los Angeles, California 90095, USA. [3] Department of Chemical and Biomolecular Engineering, University of California, Los Angeles, California 90095, USA. Correspondence and requests for materials should be addressed to K.W. (email: kenji55@u-shizuoka-ken.ac.jp)

Quinoline and quinolone alkaloids are found in diverse types of organisms, and those secondary metabolites exhibit a variety of useful biological activities, including antibacterial, antimalarial, antiviral and antitumor activities[1]. Thus, the quinolone motif found commonly among such alkaloids is used as a versatile scaffold for preparing libraries of bioactive compounds[2]. 4′-methoxyviridicatin 3 (Fig. 1a), described in our previous report[3], and related viridicatin 6 (Fig. 1b) produced by various Penicillium sp.[4,5] carry a structurally and medicinally interesting viridicatin scaffold[6] that is also found in other quinolone and quinolinone alkaloids[7–11]. Our previous investigation of the aspoquinolone/penigequinolone biosynthetic pathways (Fig. 1a, aspoquinolone and penigequinolone) has revealed a number of unique mechanisms involved in the formation of the family of natural products, including a highly unconventional dehydrogenation-mediated elongation of a prenyl chain[12] and subsequent cationic epoxide rearrangements of the hydroxylated prenyl chain[13] that generate structurally diverse side chain groups onto the viridicatin scaffold of 3. We have also shown[3] that the bimodular nonribosomal peptide synthetase (NRPS) AsqK catalyzes the condensation of anthranilic acid with different amino acids to form cyclopeptins, the 6, 7-bicyclic precursor of the viridicatin scaffold. The amino acid can be L-phenylalanine to form cyclopeptin 4 or O-methyl-L-tyrosine to form (−)-4′-methoxycyclopeptin 1. Subsequently, the non-heme α-ketoglutarate-dependent dioxygenase AsqJ singlehandedly performs a sequential iron-catalyzed desaturation and epoxidation of cyclopeptins to produce cyclopenins[14–16], the key intermediate in the formation of the 6,6-quinolone viridicatin scaffold. In case of (−)-4′-methoxycyclopenin 2, a spontaneous nonenzymatic rearrangement transforms 2 into 3. However, the inability of non-4′-methoxylated (−)-cyclopenin 5 to undergo spontaneous conversion to 6 has suggested the involvement of another enzyme that catalyzes the 6,7-benzodiazepinedione-to-6, 6-quinolone conversion[3]. This is in agreement with previous reports of an enzyme named cyclopenase that was reported to be present in a fungal cell extract that converted 5 to 6[17–21].

Here, we report the identification of AsqI as the elusive cyclopenase that has eluded isolation and detailed characterization to date. Biochemical analysis reveals AsqI as a metalloprotein that requires zinc for its activity. X-ray crystallographic studies and further in vitro assays of AsqI and its mutants, along with computational investigations of the reaction pathways for the conversion of cyclopenins to viridicatins reveals the mechanism through which the ring-contraction transformation is accomplished.

## Results and Discussion

**Hemocyanin-like zinc-binding proteins as the cyclopenase.** Analysis of the Aspergillus nidulans aspoquinolone (asq) biosynthetic gene cluster[3], the closely related Penicillium thymicola penigequinolone (pen) biosynthetic gene cluster[12] and another related[10] Penicillium sp. FKI-2140 penigequinolone (png) biosynthetic gene cluster reported here (Supplementary Fig. 1 and Supplementary Table 1) identified a gene with unknown function that was homologous to hemocyanin, a copper-containing oxygen transporter, in all three clusters (Supplementary Fig. 2). To examine the activity of AsqI and PngL in detail, their genes were cloned and expressed as a hexahistidine-tagged protein in Escherichia coli (Supplementary Figs. 3 and 4 and Supplementary Methods). When the recombinant AsqI was incubated with 5, a rapid formation of 6 was observed (Fig. 2a), confirming that AsqI is indeed the missing cyclopenase. However, the recombinant PngL failed to convert 5 to 6. Homology to hemocyanin suggested involvement of metal ions in substrate binding and catalysis by

those enzymes. When AsqI was treated with ethylenediaminetetraacetic acid, it completely lost its cyclopenase activity (Supplementary Fig. 5). However, re-introduction of different metal ions ($Fe^{2+}$, $Fe^{3+}$, $Co^{2+}$, $Mn^{2+}$, $Ni^{2+}$, $Cu^{2+}$, and $Zn^{2+}$) led to a varying degree of recovery of the activity, with zinc ion achieving the most outstanding recovery.

The activities of AsqI and PngL were examined further by steady-state kinetic analyses using 5 and 2 as substrates (see Supplementary Methods for PngL analysis). The kinetic parameters for AsqI with 5 were $k_{cat}$ 8.02 ± 0.27 min$^{-1}$ and $K_m$ 0.068 ± 0.0089 mM (Supplementary Fig. 6a). Supplementing the reaction mixture with 30 μM $ZnCl_2$ doubled $k_{cat}$ to 16.0 ± 0.30 min$^{-1}$ but had a little effect on $K_m$, which was 0.066 ± 0.0052 mM, indicating the important catalytic role the metal ion plays (Supplementary Fig. 6b). On the other hand, the kinetic parameters for AsqI with 2 were $k_{cat}$ 13,600 ± 2030 min$^{-1}$ and $K_m$ 2.86 ± 0.52 mM (Supplementary Fig. 7). Comparison of the kinetic parameters suggests that AsqI has a substantially higher affinity toward 5 than the 4′-methoxylated counterpart. Lower $k_{cat}$ for 5 is likely a reflection of the inherent difficulty of the ring-contraction transformation on the 4′-unsubstituted substrate, while the dramatically higher $k_{cat}$ value for 2 is due to the electron-donating p-methoxy group that facilitates the transformation. This point will be discussed further below. Difficulty in purifying the recombinant PngL prevented determination of accurate kinetic parameters for PngL with 2. However, $V_{max}$ was estimated to be 140 μM min$^{-1}$ and $K_m$ to be 0.35 ± 0.030 mM (Supplementary Fig. 8), making PngL similar to AsqI in its catalytic ability to convert 2 to 3 efficiently. The strict specificity of PngL for the 4′-methoxylated substrates is consistent with the findings that Penicillium sp. FKI-2140 produces only 4′-methoxylated viridicatins, such as penigequinolones and yaequinolones[10], whereas A. nidulans produces not only 4′-methoxylated viridicatins but also non-4′-methoxylated viridicatins, such as aflaquinolones and aniduquinolones[11]. Moderate differences in the amino acid sequence between AsqI and PngL (57% identity and 71% similarity based on the amino acid sequence alignment performed using EMBOSS Matcher[22], Supplementary Fig. 9) could account for the difference in their substrate specificities. Furthermore, our proposed reaction scheme also predicts elimination of methyl isocyanate 7 upon ring contraction (Fig. 1b). Since methyl isocyanate can react with water and easily decompose into methylamine and carbon dioxide, thiophenol 8, a stronger nucleophile than water, was included in the reaction mixture to trap 7 in the form of carbamothioate 9 for detection (Fig. 1c). The result of in vitro assay on AsqI clearly indicated the concurrent formation of 6 and 9 (Fig. 2b, Supplementary Figs. 10–15, Supplementary Table 3 and Supplementary Methods). Similarly, the proposed ring rearrangement places the distal carbon of the exocyclic epoxide C10 of 5 at the C4 position of the bicyclic system of 6 (Fig. 1b). Chemical characterization of 6 (Supplementary Figs. 16–21 and Supplementary Table 4) isolated from feeding experiment using 5 labeled with $^{13}C$ at the C10 position (Supplementary Figs. 22–24 and Supplementary Methods) confirmed the presence of the labeled carbon at the C4 position of 6 (Supplementary Figs. 25–27 and Supplementary Methods), providing a strong support for the proposed mechanism of the AsqI-catalyzed ring-contraction transformation.

**Structural and computational analysis of cyclopenases.** Next, to establish the active-site architecture, the crystal structure of AsqI was determined (Fig. 3a, Table 1 and Supplementary Figs. 28–30). Based on the structure and the sequence homology of AsqI to hemocyanin (Supplementary Fig. 2), residues His176, His180,

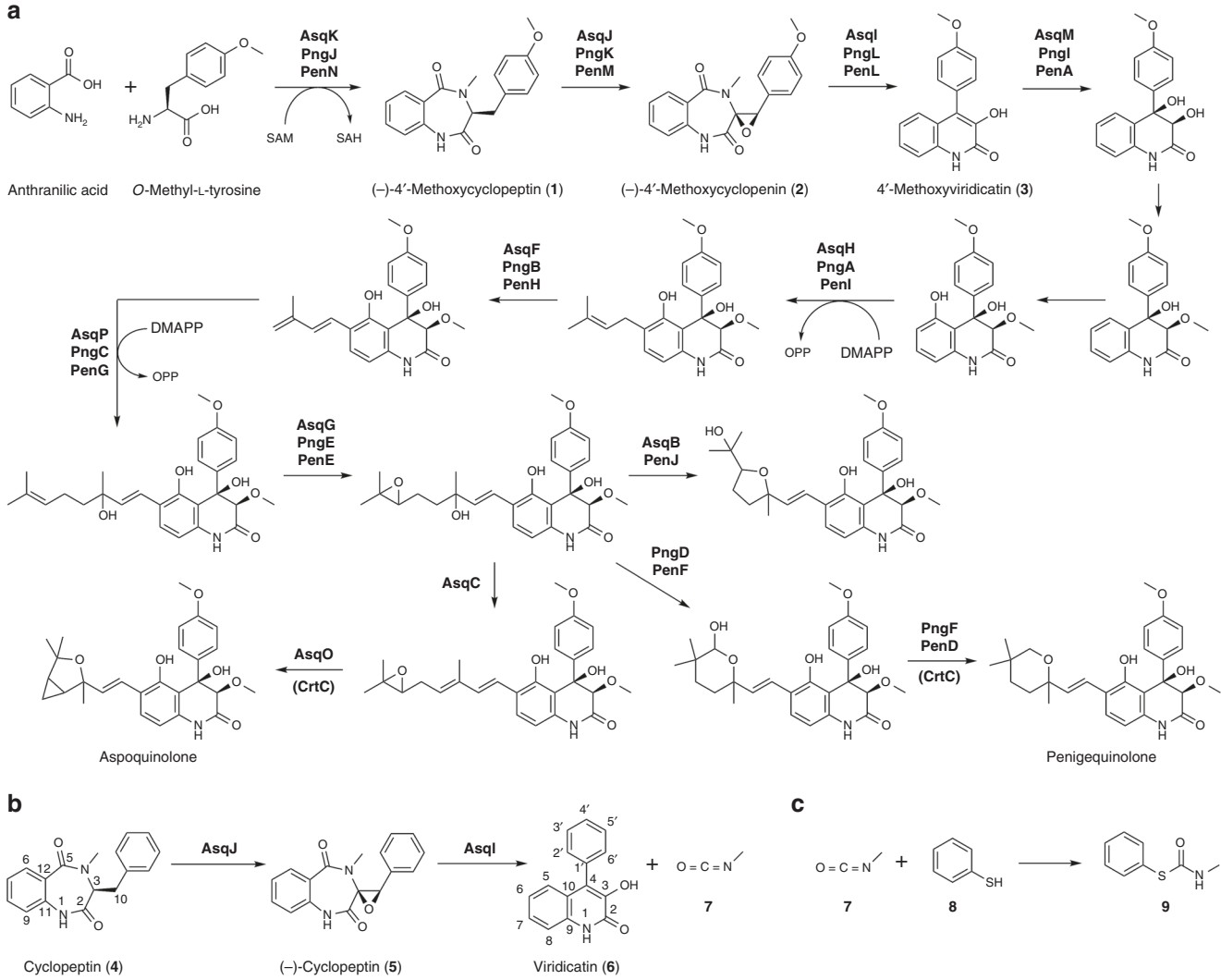

**Fig. 1** Biosynthesis of viridicatin-type fungal alkaloids. **a** Proposed *asq* biosynthetic pathway in *Aspergillus nidulans* for the formation of aspoquinolone, and proposed *pen/png* biosynthetic pathway in *Penicillium* sp. for the formation of penigequinolone. **b** Proposed transformation of the non-4′-methoxylated 6,7-bicyclic cyclopeptin **4** to 6,6-bicyclic viridicatin **6**. The non-heme dioxygenase AsqJ catalyzes the epoxidation of **4** to yield (−)-cyclopenin **5**. Then, the hemocyanin-like AsqI catalyzes the ring contraction in **5** to form **6** via a 6-*endo-tet* cyclization. **c** The product methyl isocyanate **7** was trapped by converting it to carbamothioate **9** with thiophenol **8**. DMAPP, dimethylallyl pyrophosphate; OPP, diphosphate

and His208 were identified to form the metal-binding site A in AsqI. However, the second tri-histidine metal-binding site B found in hemocyanins was formed by His346, Leu350, and Y387 in AsqI and no metal ion was observed there. While His346Ala mutation did not reduce the activity of AsqI, alanine mutation of each of the site-A histidine residues resulted in a nearly complete loss of activity, suggesting that the site-A metal plays a crucial role in substrate binding and catalysis (Fig. 3b). Mutagenesis of the residues Arg184, Asp322, and Asn347 near the metal-binding site also resulted in a substantial activity loss, indicating their involvement in catalysis. Since AsqI failed to crystallize in the presence of the substrates or the products, details of how AsqI interacts with the ligands could not be elucidated. Thus, computational analyses of the reaction were carried out to gain further insight into this interesting transformation.

Calculation of the proposed reaction pathway for the formation of **6** in the absence of any acidic or basic catalysts predicts a barrier of activation of 44.5 kcal mol$^{-1}$, indicating conclusively that this reaction practically does not proceed at room temperature (Supplementary Fig. 31 and Supplementary Data 1). Calculations with methylammonium as a model for acid catalysis

result in free energy pathways with three transition states for the formation of both **6** and its 4′-methoxylated counterpart **3** (Supplementary Fig. 32 and Supplementary Data 2 and 3). In this case, the rate-determining steps of the reaction have a barrier of 30.8 and 24.3 kcal mol$^{-1}$, for **6** and **3**, respectively. The latter barrier is accessible at room temperature, and explains the experimental observation that **3** is far more readily formed than **6** and can be obtained from **2** simply in the presence of weak acids. The reaction is more facile for the 4′-methoxylated compound, because the electron-donating ability of the methoxy group stabilizes the transient benzylic carbocation that is formed when the epoxide is opened prior to the attack of the aromatic nucleophile. To account for the role of zinc and the enzyme in the reaction of **5**, we also optimized the structure of **5** in complex with Zn$^{2+}$ and two imidazole molecules, which model the two histidine side chain groups identified experimentally as the metal ligands. The structure (Supplementary Fig. 33 and Supplementary Data 4) shows that the metal ion is coordinated to the imidazole molecules and the epoxide and C2 amide oxygen atoms of the substrate in a tetrahedral arrangement as frequently observed in zinc-complexed protein structures[23]. The free energy pathway

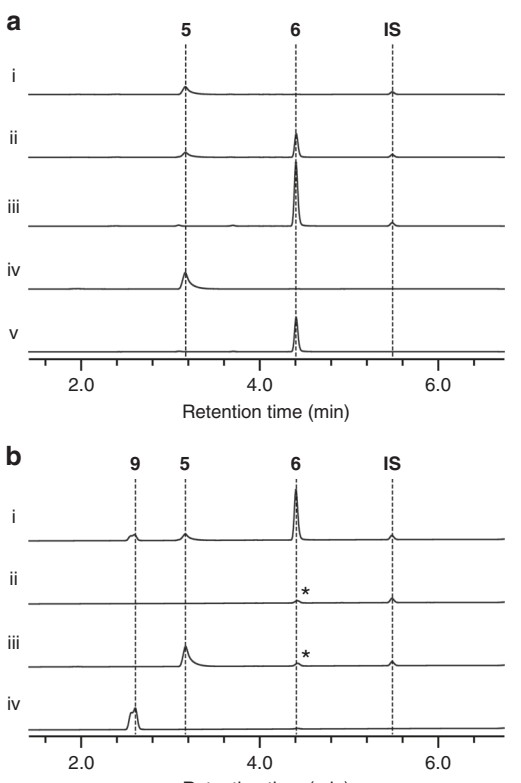

**Fig. 2** In vitro analyses of the activity of AsqI against **5**. Detailed reaction conditions are described in the Methods section. **a** In vitro assays of AsqI with **5** as a substrate. HPLC profile of the reaction mixture containing (i) heat-inactivated AsqI with **5** as a negative control; (ii) active AsqI with **5**, showing the formation of **6** after 5 min of incubation; (iii) active AsqI with **5**, showing a complete conversion of **5** to **6** after 30 min of incubation; (iv) the authentic reference of **5**; and (v) the authentic reference of **6**. Traces were monitored at 280 nm. **b** In vitro assays of AsqI with **5** and **8**. HPLC profiles of the reaction mixture containing (i) active AsqI with **5** and **8**, showing the formation of **6** and **9** after 30 min of incubation; (ii) AsqI and **8** without **5** after 30 min of incubation as a negative control; (iii) **5** and **8** incubated for 30 min without AsqI as a negative control; and (iv) the authentic reference of **9**. Traces were monitored at 244 nm. These peaks denoted by asterisks are derived from **8**

computed for the formation of **6** from this $Zn^{2+}$-coordinate complex shows a two-step reaction, where the nucleophilic attack of the aromatic ring on the epoxide (Fig. 4a TS1) has an activation barrier of only 13.5 kcal mol$^{-1}$, much smaller than in the case of acid catalysis (see above). The second step, elimination of **7** (Fig. 4a TS2), is almost barrierless (0.7 kcal mol$^{-1}$). This is consistent with the experimental observation of the fast rate of AsqI-catalyzed formation of **6**. For the enzyme-catalyzed reaction, a $Zn^{2+}$ cation in the active site, which is tetrahedrally coordinated to the side chains of His176 and His180 and the epoxide and C2 carbonyl oxygen of the substrate, is proposed to act as the Lewis acid to initiate the unusual ring-contraction reaction (Fig. 4b).

Lastly, to understand the substrate specificity of the cyclopenase-type catalysts, a homology model of PngL was constructed based on the crystal structures of AsqI and an arthropod phenoloxidase[24] (Supplementary Methods). The most significant difference between AsqI and PngL/PenL is that the latter has the metal-binding site B rather than the site A intact (Supplementary Fig. 34). Detailed kinetic characterization of PngL could not be performed due to poor expression of PngL

mutants (Supplementary Fig. 4). However, the structural models and the sequence differences of the active site-lining residues suggest that the substrate-binding modes of AsqI and PngL differ substantially. The moderate activity against **2** and no activity against **5** suggest that PngL may rely on the inherent reactiveness of the 4′-methoxylated substrate and drive the reaction mostly by pre-organizing the substrate into a reactive conformation rather than metal-based catalysis.

Our experimental results clearly indicate that the hemocyanin-like protein AsqI is the cyclopenase, which is the enzyme in the viridicatin biosynthetic pathway that has eluded isolation for nearly four decades. We have also discovered PngL, a homolog of AsqI from a *Penicillium* sp. that only accepts 4′-methoxylated compound as its substrate. A series of biochemical and computational characterizations conducted in this study support the proposed catalytic mechanism of the highly unusual ring-contraction transformation (Fig. 4b). In this reaction, the active-site zinc ion acts as a Lewis acid catalyst to activate the substrate epoxide and facilitate essentially an anti-Baldwin-type epoxide-opening 6-*endo-tet* cyclization[25]. Subsequently, the conversion of a 6,7-bicyclic skeleton into a 6,6-bicyclic quinolone framework occurs upon elimination of **7** and the following keto–enol tautomerization results in the formation of the product. The difference in the substrate specificity between AsqI and PngL may arise from the difference in the active-site architecture that alters the effectiveness of how the zinc ion engages the bound substrate. Through the current and previous studies, it has become clear that the key to the viridicatin framework formation is the dioxygenase AsqJ-catalyzed epoxidation of benzodiazepine-diones[3] and the subsequent ring contraction by the hemocyanin-like AsqI. These findings hint the potential of using the catalytic sequence powered by the combined

## Table 1 Data collection and refinement statistics

| | Apo AsqI (native) | Apo AsqI (SAD) | AsqI–zinc complex |
|---|---|---|---|
| **Data collection** | | | |
| Space group | *I*222 | *I*222 | *I*222 |
| Cell dimensions | | | |
| *a*, *b*, *c* (Å) | 85.0, 117.4, 159.0 | 84.4, 113.5, 158.2 | 84.5, 114.8, 157.7 |
| Resolution (Å) | 19.9–2.30 (2.38–2.30) | 20.0–3.51 (3.71–3.51) | 19.8–2.91 (3.01–2.91) |
| $R_{merge}$ | 0.106 (0.831) | 0.321 (0.798) | 0.145 (0.704) |
| $I/\sigma I$ | 15.22 (2.33) | 8.43 (3.24) | 12.96 (2.44) |
| Completeness (%) | 98.9 (97.2) | 99.3 (99.3) | 99.9 (99.2) |
| Redundancy | 6.7 | 14.2 | 6.7 |
| **Refinement** | | | |
| Resolution (Å) | 19.9–2.30 | | 19.8–2.91 |
| No. reflections | 35,077 | | 17,156 |
| $R_{work}/R_{free}$ | 19.28 / 21.99 | | 19.70 / 21.53 |
| No. atoms | | | |
| Protein | 4664 | | 4603 |
| Ligand/ion | 0 | | 1 |
| Water | 279 | | 1 |
| B-factors | | | |
| Protein | 33.90 | | 33.30 |
| Ligand/ion | 36.30 | | 38.60 |
| Water | | | 22.10 |
| R.m.s. deviations | | | |
| Bond lengths (Å) | 0.009 | | 0.014 |
| Bond angles (°) | 1.145 | | 1.236 |

One crystal was used for each structure. Values in parentheses are for highest-resolution shell

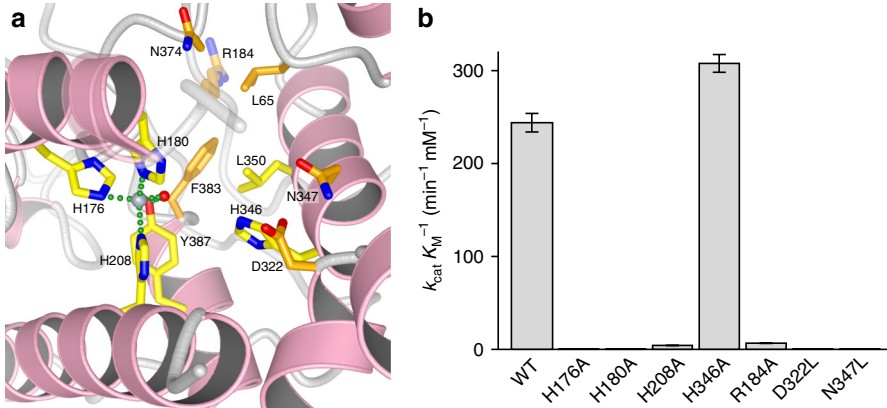

**Fig. 3** The structure determination and mutagenesis of AsqI. **a** The metal-binding site of AsqI. Carbon atoms of the stick models of the residues corresponding to the conserved metal-binding residues are in lighter yellow, while carbon atoms in other residues are in darker yellow. Oxygen and nitrogen atoms are in red and blue, respectively, and α-helices are colored in pink. The bound zinc ion and water molecule are shown as a blue-gray and red sphere, respectively. Bonds between the zinc atom and other bound atoms are shown as green dotted lines. The semi-transparent apo AsqI structure is overlaid to indicate the position of Arg184, which is disordered in the AsqI–zinc complex structure. **b** Comparison of the catalytic abilities of the wild-type (WT) AsqI and its mutants to transform **5** into **6**. The measurement for each mutant is a mean of triplicate measurements. The standard deviation is given in the plot as an error bar at the top of the bar

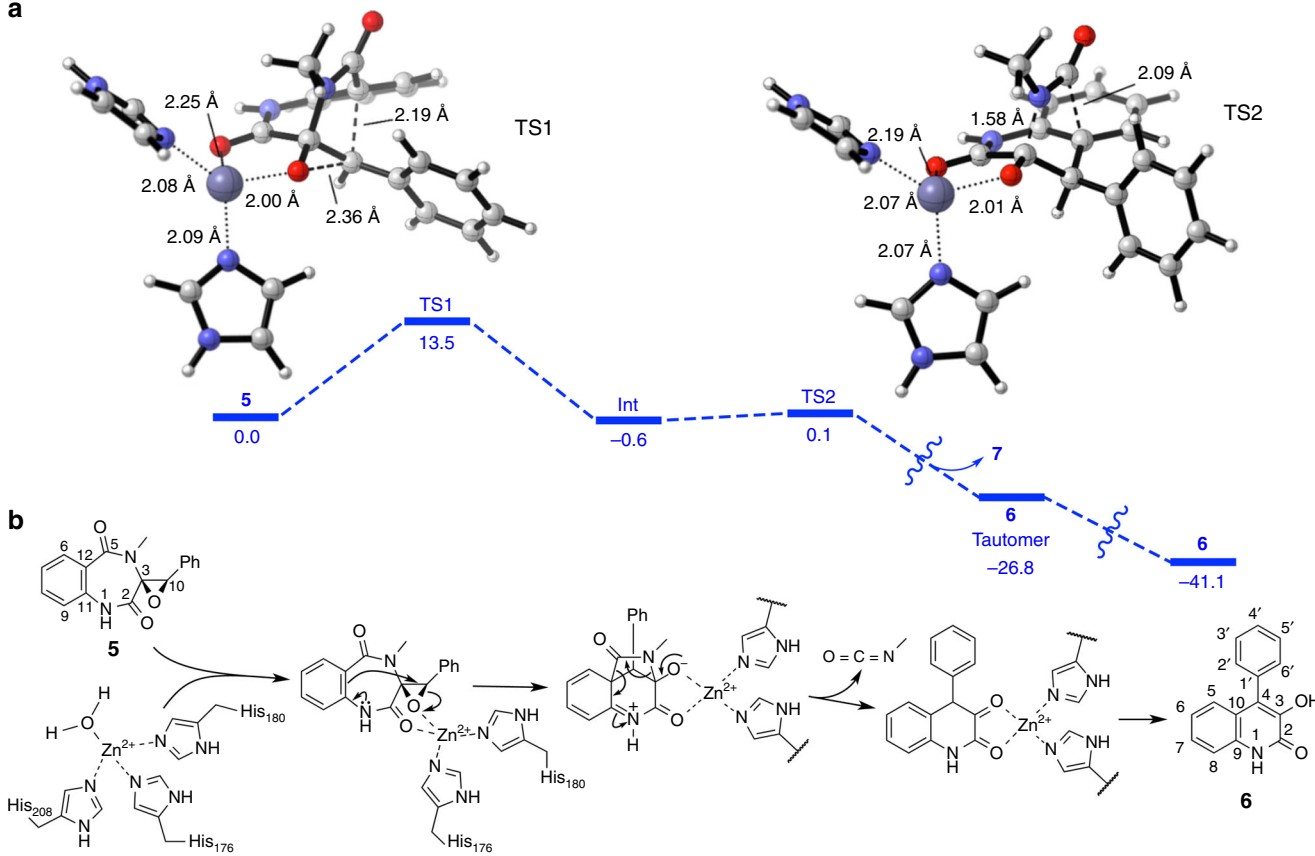

**Fig. 4** Computational analyses of the AsqI-catalyzed ring contraction reaction. **a** The metal-binding site of AsqI. Computational results on the full reaction pathway for the transformation of **5** into **6**, catalyzed by a $Zn^{2+}$ ion tetracoordinated to the substrate and two molecules of imidazole as a model for a histidine side chain. Calculated free energy differences ($\Delta G$) in kcal mol$^{-1}$ are given below the bold bars. The largest barrier is the first transition state (TS1) involving the opening of the cyclopenin epoxide and formation of the methylamide-bridged intermediate (Int). The second transition state (TS2) leading to the elimination of **7** and formation of the 6,6-bicyclic core is virtually barrierless. Computed TS1 and TS2 structures are shown. Carbon, oxygen, nitrogen and zinc atoms in the ball-and-stick models are in white, red, darker blue, and lighter blue, respectively. Breaking and forming bonds are shown as thick broken lines, while bonds between the zinc atom and other bound atoms are shown as thin dotted lines. **b** The proposed mechanism for the conversion of **5** to **6** catalyzed by the AsqI–$Zn^{2+}$ complex. The $Zn^{2+}$ ion bound to His176 and His180 is thought to catalyze the reaction as a Lewis acid

dioxygenase–hemocyanin activity as a unique biosynthetic or chemoenzymatic approach in generating various 6, 6-quinolones from benzodiazepinediones prepared from anthranilic acid with various different amino acids.

## Methods

**In vitro analysis of the activity of AsqI against 5.** The assay mixture (40 µL) containing 2 µM AsqI and 0.4 mM of (–)-cyclopenin **5** in MES Na buffer (100 mM 2-morpholinoethanesulfonic acid sodium salt (MES Na), 100 mM NaCl, pH 5.5) was incubated at 30 °C for 5 or 30 min. Heat-inactivated samples of AsqI was used in the indicated reaction as a negative control. After a 30 min incubation, the reaction was quenched by addition of 80 µL of ethyl acetate (EtOAc) containing 10 µM of anthraquinone as an internal standard (IS). The organic layer was separated by centrifugation, and the isolated organic fraction was dried in vacuo. The dried material was dissolved in 60 µL of $N,N$-dimethylformamide (DMF) and subjected to LC–MS analysis performed with a Thermo SCIENTIFIC Q-Exactive liquid chromatography mass spectrometer using both positive and negative electrospray ionization. LC was performed using an ACQUITY UPLC 1.8 µm, 2.1 × 50 mm C18 reversed-phase column (Waters) and separated on a linear gradient of 10–50% (v v$^{-1}$) acetonitrile (CH$_3$CN) in H$_2$O supplemented with 0.05% (v v$^{-1}$) formic acid at a flow rate of 500 µL min$^{-1}$. Peak heights of different samples were standardized by scaling the heights of the IS peaks in all samples.

**Kinetic analysis of AsqI.** Different concentrations of **5** (0.05, 0.1, 0.2, 0.4 and 0.8 mM) was mixed with 2.0 µM of AsqI in MES Na buffer and with or without 30 µM of ZnCl$_2$ in a total reaction volume of 40 µL. After 1 or 2 min of incubation at 30 °C, the reaction was quenched with 80 µL of EtOAc containing 10 µM of anthraquinone as an IS. The organic layer was separated by centrifugation, and the isolated organic fraction was dried in vacuo. The dried material was subjected to LC–MS analysis as described earlier. Initial reaction rates were determined on the basis of the amount of **6** present in the sample, and the data points were plotted as shown in Supplementary Fig. 6. For the reaction with (–)-4′-methoxycyclopenin **2** as a substrate, different concentrations of **2** (0.05, 0.1, 0.2, 0.4 and 0.8 mM) was mixed with 0.05 µM of AsqI in MES Na buffer in a total reaction volume of 40 µL. After 1 min of incubation at 30 °C, the reaction was quenched with 80 µL of EtOAc containing IS (10 µM anthraquinone). The organic layer was separated by centrifugation, and the isolated organic fraction was dried in vacuo. The dried material was subjected to LC–MS analysis as described earlier. Initial reaction rates were determined on the basis of the amount of **3** present in the sample, and the data points were plotted as shown in Supplementary Fig. 7. Kinetic parameters were calculated by nonlinear regression of the data using GraphPad Prism software (GraphPad Software, Inc.). Each data point is a mean of triplicate measurements. The standard deviation is given in the plot as an error bar at each data point.

**Preparation of the selenomethionine derivative of AsqI.** BL21(DE3) harboring plasmid pKW18244 (Supplementary Fig. 3) was grown overnight in 50 mL of LB medium with 50 µg mL$^{-1}$ kanamycin at 37 °C. Five liters of fresh M9 medium with 50 µg mL$^{-1}$ kanamycin was inoculated with 50 mL of the overnight culture and incubated at 37 °C. Amino acid mixture (100 µg mL$^{-1}$ L-lysine monohydrochloride, 100 µg mL$^{-1}$ L-threonine, 100 µg mL$^{-1}$ L-phenylalanine, 50 µg mL$^{-1}$ L-isoleucine, 50 µg mL$^{-1}$ L-leucine, 50 µg mL$^{-1}$ L-valine, and 50 µg mL$^{-1}$ seleno-L-methionine (Se-Met)) were added to the culture when the optical density at 600 nm (OD$_{600}$) was 0.3 and incubation was continued until OD$_{600}$ reached 0.6. Then expression of the gene was induced with 200 µM isopropylthio-β-D-galactoside (IPTG) at 18 °C. Incubation was continued for another 24 h, after which cells were harvested by centrifugation at 10,000×$g$ for 5 min. Cell disruption and purification of the protein were performed in the same manner as described for non-labeled AsqI.

**Structure refinement, model completion and analysis.** X-ray diffraction data from the crystals of the apo and zinc-complexed AsqI as well as apo Se-Met derivative were collected as described in Supplementary Methods. The initial structure of the Se-Met derivative of AsqI was determined by the single-wavelength anomalous dispersion (SAD) method using the PHENIX AutoSol wizard[26]. The structure model was built with the program COOT[27] and refined with phenix.refine[28]. Using the SAD-derived structure as a search model, native structures, were determined by molecular replacement using the program Phaser[29]. The models were built with COOT[27] and refined with phenix.refine[28]. Data collection and refinement statistics are shown in Table 1. The Ramachandran statistics for the native apo AsqI model indicated that 97.9% and 2.1% of the residues were in the favored and the allowed regions, respectively, with none in the disallowed region. Similarly for the zinc-bound AsqI model, 95.5% and 4.5% of the residues were in the favored and the allowed regions, respectively, with none in the disallowed region. The least-squares superimposition of the apo structure model with the zinc-complex structure model was performed (root-mean-square deviation of 0.378 Å for all the matching Cα atoms) for structural analysis of AsqI.

**Computational analysis.** Quantum mechanical calculations by density functional theory (DFT) were performed using Gaussian 09 (Revision D.01)[30]. For the uncatalyzed and methylammonium-catalyzed pathways, geometries were optimized using M06-2×[31] with the 6-31G(d) basis set. For the Zn$^{2+}$ pathway, geometries were optimized at the B3LYP/6-31G(d)/LANL2DZ(Zn) level with the SMD solvation model[32] (Et$_2$O, $\varepsilon = 4$). Verification of whether the geometries are minima (zero imaginary frequencies) or transition structures (TS, one imaginary frequency) is accomplished by normal mode vibrational analysis on the stationary points. All TS were further analyzed by IRC calculations to confirm that they connect the expected minima. A standard state of 1 atmosphere of pressure and 298 K were used to obtain ZPE, enthalpy and free energy corrections. Truhlar's quasiharmonic oscillator approximation was used to compute free energies, where all frequencies below 100 cm$^{-1}$ were set to 100 cm$^{-1}$[33,34]. In all cases, single point energies were obtained at the M06-2× /6-311 + G(d,p)SMD($\varepsilon = 4$) level of theory[31]. The resulting energies were used to correct those obtained from the optimizations[35]. Computed structures are illustrated with CYLView[36]. All of the coordinates of the computed structures are given in Supplementary Data 1–4.

**Data availability.** The coordinates, proven to have good stereochemistry from the Ramachandran plots, were deposited at the RCSB Protein Data Bank under the accession codes 5YY3 (apo AsqI) and 5YY2 (AsqI–zinc complex). All other data are available from the authors upon reasonable request.

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

## Acknowledgements

This work was supported by the NIH (1DP1GM106413 and 1R35GM118056 to Y.T.), the NSF (CHE-1361104 to K.N.H.) and JSPS Program for Advancing Strategic International Networks to Accelerate the Circulation of Talented Researchers (No. G2604 to K.W.). This work was also supported in part by the Japan Society for the Promotion of Science (JSPS) (K.W., 15KT0068, 26560450), Innovative Areas from MEXT, Japan (K.W., 16H06449), the Institution of Fermentation at Osaka (K.W.), the Takeda Science Foundation (K.W.), the Uehara Memorial Foundation (K.W.) and the Princess Takamatsu Cancer Research Fund (K.W.). P.A.C acknowledges the Fonds de recherche du Québec-Nature et Technologies (FRQNT) for a postdoctoral fellowship. Computations were performed on the Hoffman2 cluster at UCLA. We acknowledge the kind support of the beamline staff of Photon Factory for X-ray data collection.

## Author contributions

S.K., K.H., H.H., Y.H. and K.W. conceived and designed the study. S.K. designed and performed molecular cloning. S.K. performed the heterologous expression and purification as well as in vitro characterization of the enzymes. K.H. and H.H. performed crystallographic study and analyzed metal ions in AsqI. S.K. and Y.H. elucidated the chemical structures. P.A.C. and K.N.H. performed the computational experiments. S.K. and Y.H. analyzed DNA and amino acid sequence analysis. All authors analyzed and discussed the results. K.N.H., Y.T. and K.W. prepared the manuscript.
