## [Peer Review File · Nature Communications]

Reviewers' comments:

Reviewer #1 (Remarks to the Author):

Watanabe and coworkers report on a mechanistically very interesting and surprising enzyme that is involved in fungal quinolone biosynthesis. The study combines various aspects including a thorough biochemical characterisation of the enzyme, the enzyme structure and quantum chemical calculations on the catalysed reaction. The trapping of the coupled product methylisocyanate is particularly elegant. The manuscript is nicely written and follows a good logic approach. The discussion about the chemistry is very strong, e. g. when reactivities of the methoxy and the non-substituted substrate are compared. This fine manuscript will be of high interest to natural product chemists and other broadly interested colleagues and thus deserves publication in Nat. Commun.

A few minor points should be addressed:

1. Please explain the biosynthetic pathway of Figure 1 in more detail in the introduction. This will be interesting for readers who are not familiar with the compound class and its biosynthesis. Especially the epoxidation from 5 to 6 is remarkable, is anything known about the mechanism?
2. The sentence "However, the inability of AsqJ to convert non-4'-methoxylated cyclophenin 4 to 2 suggested the involvement of an enzyme in the 6,7-benzodiazepinedione-to-6,6-quinolone conversion³." should be rephrased. It is not logic to state that an enzyme must be involved, just because AsqJ failed to catalyse the discussed conversion. Perhaps the authors wanted to write "...another enzyme...".
3. In Figure 1, compound numbers should increase in the order of appearance of the compounds. What is the difference between 7a and 7b (and 8a and 8b)?
4. The kinetic data for substrates 4 and 6 are very different. The authors give a good explanation for the different k_{cat} . Does the enzyme structure give any explanation why the K_m is so different?
5. Instead of "non-4'-methoxylated viridicatin" the term "4'-nonsubstituted viridicatin" should be preferred.
6. Please place all Tables, Figures etc. in the SI in the order of their discussion in the main text / in the order of the workflow.
7. The sentence "The structure (Supplementary Fig. 35) shows that the metal ion is coordinated to the imidazole molecules and the epoxide and C2 amide oxygen atoms of the substrate in a tetrahedral arrangement, which is also the preferred coordination structure for Zn^{2+} in proteins" should be rephrased, because for Zn^{2+} (electron configuration = $3d^{10}$) a tetrahedral coordination is generally preferred, not just in proteins.
8. In Figure 4 in the central structure (before cleavage of methyl isocyanate) one of the mechanistic arrows is not correct. The arrow starting at the alcoholate should show formation of a C=O double bond, and the electrons of the C-N bond should migrate to form the C=N double bond in methyl isocyanate.

Reviewer #2 (Remarks to the Author):

The manuscript by Watanabe and coworkers describes the functional evaluation of AsqI, a key

enzyme in viridicatin biosynthesis that acts as a so far elusive cyclophenase. The authors characterize AsqI and the homologous PngL in vitro using native substrates, providing kinetic data for both these enzymes. The authors also provide insights into the catalytic mechanism of these enzymes by a combination of in vitro assays, mutational studies and computational analyses. All experiments are convincing and the results are presented in a clear and well-written manuscript. Overall, this work thus provides important new insights into viridicatin biosynthesis that are of interest to natural product chemists and enzymologist alike. I thus support publication of this work practically as is. I would only encourage to include a broader discussion of previous work, also by other labs (such as the recent excellent Angew. Chem. and Nat. Commun. papers on AsqJ by the Groll and Kailla labs), in the introductory section of the paper.

Reviewer #3 (Remarks to the Author):

The X-ray structures presented in this paper are of good quality. I believe they accurately fit the diffraction data and conclusions based on them can be trusted. There are a number of minor points I would like to bring to the attention of the authors and the editor. I do not consider these matters to be critical flaws and believe the conclusions of the paper are sound.

Page -7 in Supplementary Methods: What is 30% in the reservoir based on the condition of Morpheus 2-46 ? Concentration of the zinc for crystallization of AsqI-zinc complex is not shown. How the authors prepare the zinc-bound form ?

X-ray data: Supp Table 3 in page 22-23: R-meas and CC(1/2) values are more standard index for the data collection statistics. They should be included rather than R-merge. "Refined reflections" is not correct. Reflection cannot be refined. It should be changed to "Work reflections" to form a pair with "Free reflections".

Fig S11 A, All helices should be labeled.

Fig S11B: Distances of Zn-His coordination should be shown in the figure.

Fig S12: The structural differences of the zinc-binding site is very difficult to understand. Addition of another panel focusing on the zinc site would be helpful.

Phasing: The description about the phasing method is not enough in this manuscript. For example, it is described that SAD with PHENIX AutoSOI wizard was used. However, only Supplementary Table 3 tells us Se was used as an atom of anomalous dispersion. Moreover, there is little information on the expression, purification and crystallization of the Se-Met derivative.

Response to Reviewers

Reviewer #1 (Remarks to the Author):

Watanabe and coworkers report on a mechanistically very interesting and surprising enzyme that is involved in fungal quinolone biosynthesis. The study combines various aspects including a thorough biochemical characterisation of the enzyme, the enzyme structure and quantum chemical calculations on the catalysed reaction. The trapping of the coupled product methylisocyanate is particularly elegant. The manuscript is nicely written and follows a good logic approach. The discussion about the chemistry is very strong, e. g. when reactivities of the methoxy and the non-substituted substrate are compared. This fine manuscript will be of high interest to natural product chemists and other broadly interested colleagues and thus deserves publication in Nat. Commun.

A few minor points should be addressed:

1. Please explain the biosynthetic pathway of Figure 1 in more detail in the introduction. This will be interesting for readers who are not familiar with the compound class and its biosynthesis. Especially the epoxidation from 5 to 6 is remarkable, is anything known about the mechanism?

Further details are added to the revised manuscript. (Main text, page 2, bottom half)

2. The sentence "However, the inability of AsqJ to convert non-4'-methoxylated cyclophenin 4 to 2 suggested the involvement of an enzyme in the 6,7-benzodiazepinedione-to-6,6-quinolone conversion³." should be rephrased. It is not logic to state that an enzyme must be involved, just because AsqJ failed to catalyse the discussed conversion. Perhaps the authors wanted to write "...another enzyme...".

This sentence is rephrased as suggested. (Main text, page 2, 2nd line from last)

3. In Figure 1, compound numbers should increase in the order of appearance of the compounds. What is the difference between 7a and 7b (and 8a and 8b)?

Compound numbering is corrected. However, we would like to maintain the order to be the order of appearance in the text, not in the figure.

Difference between **a** and **b** has to do with the stereochemistry of the ring. Because this detail is not directly relevant to the focus of the current manuscript, we have eliminated the designations.

4. The kinetic data for substrates 4 and 6 are very different. The authors give a good explanation for the different k_{cat} . Does the enzyme structure give any explanation why the K_m is so different?

Unfortunately, we were unable to obtain a crystal of AsqI in complex with any substrates or products. As such, it is very difficult to come up with sound structural explanations for the results from our kinetics characterizations.

5. Instead of "non-4'-methoxylated viridicatin" the term "4'-nonsubstituted viridicatin" should be preferred.

Because "4'-methoxylation" is key to the mechanistic difference in the 6,7-to-6,6 ring contraction reaction between viridicatin and 4'-methoxylated viridicatin, we wish to emphasize the presence/absence of "4'-methoxylation" by referring to the 4'-nonsubstituted viridicatin as the non-4'-methoxylated viridicatin.

6. Please place all Tables, Figures etc. in the SI in the order of their discussion in the main text / in the order of the workflow.

We have come to the current ordering as a compromise between keeping the order of the workflow in the main text and maintaining reasonable organization within the SI. As such, we would like to keep the organization of the SI as is.

7. The sentence "The structure (Supplementary Fig. 35) shows that the metal ion is coordinated to the imidazole molecules and the epoxide and C2 amide oxygen atoms of the substrate in a tetrahedral arrangement, which is also the preferred coordination structure for Zn²⁺ in proteins" should be rephrased, because for Zn²⁺ (electron configuration = 3d¹⁰) a tetrahedral coordination is generally preferred, not just in proteins.

This sentence is revised accordingly. (Main text, page 6, 2nd line from last)

8. In Figure 4 in the central structure (before cleavage of methyl isocyanate) one of the mechanistic arrows is not correct. The arrow starting at the alcoholate should show formation of a C=O double bond, and the electrons of the C-N bond should migrate to form the C=N double bond in methyl isocyanate.

The error is corrected. (Figure 4b)

Reviewer #2 (Remarks to the Author):

The manuscript by Watanabe and coworkers describes the functional evaluation of AsqI, a key enzyme in viridicatin biosynthesis that acts as a so far elusive cycloopenase. The authors characterize AsqI and the homologous PngL in vitro using native substrates, providing kinetic data for both these enzymes. The authors also provide insights into the catalytic mechanism of these enzymes by a combination of in vitro assays, mutational studies and computational analyses. All experiments are convincing and the results are presented in a clear and well-written manuscript. Overall, this work thus provides important new insights into viridicatin biosynthesis that are of interest to natural product chemists and enzymologist alike. I thus support publication of this work practically as is.

I would only encourage to include a broader discussion of previous work, also by other labs (such as the recent excellent Angew. Chem. and Nat. Commun. papers on AsqJ by the Groll and Kailla labs), in the introductory section of the paper.

Revision is made to the introduction accordingly. (Main text, page 2, bottom half)

Reviewer #3 (Remarks to the Author):

The X-ray structures presented in this paper are of good quality. I believe they accurately fit the diffraction data and conclusions based on them can be trusted. There are a number of minor points I would like to bring to the attention of the authors and the editor. I do not consider these matters to be critical flaws and believe the conclusions of the paper are sound.

Page -7 in Supplementary Methods: What is 30% in the reservoir based on the condition of Morpheus 2-46 ? Concentration of the zinc for crystallization of AsqI-zinc complex is not shown. How the authors prepare the zinc-bound form ?

30% was a typo. The condition is corrected in the revised SI. No additional zinc was provided in the crystallization solution. Zinc atom natively present in AsqI appeared to be stably bound. (SI, page 7, 9th line from last)

X-ray data: Supp Table 3 in page 22-23: R-meas and CC(1/2) values are more standard index for the data collection statistics. They should be included rather than R-merge. "Refined reflections" is not correct. Reflection cannot be refined. It should be changed to "Work reflections" to form a pair with "Free reflections".

The data table is revised accordingly. (SI, page 22–23)

Fig S11 A, All helices should be labeled.

The figure is modified accordingly. (SI, page 19)

Fig S11B: Distances of Zn-His coordination should be shown in the figure.

The information is added. (SI, page 19)

Fig S12: The structural differences of the zinc-binding site is very difficult to understand. Addition of another panel focusing on the zinc site would be helpful.

An additional panel is added. (SI, page 20)

Phasing: The description about the phasing method is not enough in this manuscript. For example, it is described that SAD with PHENIX AutoSOI wizard was used. However, only Supplementary Table 3 tells us Se was used as an atom of anomalous dispersion. Moreover, there is little information on the expression, purification and crystallization of the Se-Met derivative.

The revised description of the phasing method is expanded accordingly. (Main text, page 11, 2nd line from top)

The description of the preparation of the Se-Met derivative is added to the revised manuscript. (Main text, page 10, bottom half)

REVIEWERS' COMMENTS:

Reviewer #1 (Remarks to the Author):

All important previously raised issues have been corrected by the authors and the manuscript can now be accepted for Nat. Commun.

In their response to referees the authors state that they prefer to keep compound numbers as they are, i. e. the compound numbers increase by their appearance in the text. This is a question of personal preference and can be done like this, but the idea behind my suggestion of an increasing compound number by the order of compounds in the schemes was that the reader does not have to search within the schemes for a particular compound.

Reviewer #2 (Remarks to the Author):

The few points raised by the reviewers in the initial evaluation have properly been addressed. I thus recommend publication of this work in its current state.

Reviewer #3 (Remarks to the Author):

The manuscript by Kishimoto et al is very much improved. I fully support publication.

REVIEWERS' COMMENTS:

Reviewer #1 (Remarks to the Author):

All important previously raised issues have been corrected by the authors and the manuscript can now be accepted for Nat. Commun.

In their response to referees the authors state that they prefer to keep compound numbers as they are, i. e. the compound numbers increase by their appearance in the text. This is a question of personal preference and can be done like this, but the idea behind my suggestion of an increasing compound number by the order of compounds in the schemes was that the reader does not have to search within the schemes for a particular compound.

We decided to follow Reviewer #1's suggestion to number the compounds following their appearance in the Figure 1 pathway.

Reviewer #2 (Remarks to the Author):

The few points raised by the reviewers in the initial evaluation have properly been addressed. I thus recommend publication of this work in its current state.

Reviewer #3 (Remarks to the Author):

The manuscript by Kishimoto et al is very much improved. I fully support publication.